# Phytochemicals Controlling Enterohemorrhagic *Escherichia coli* (EHEC) Virulence—Current Knowledge of Their Mechanisms of Action

**DOI:** 10.3390/ijms26010381

**Published:** 2025-01-04

**Authors:** Patryk Strzelecki, Monika Karczewska, Agnieszka Szalewska-Pałasz, Dariusz Nowicki

**Affiliations:** Department of Bacterial Molecular Genetics, Faculty of Biology, University of Gdansk, Wita Stwosza 59, 80-308 Gdansk, Poland; patryk.strzelecki@phdstud.ug.edu.pl (P.S.); monika.szalkowska@phdstud.ug.edu.pl (M.K.)

**Keywords:** Shiga toxin, foodborne pathogens, *E. coli* O157:H7, bacteriophages, antimicrobial therapy, isothiocyanates, cinnamaldehyde, terpenes, polyphenols, biofilm

## Abstract

Enterohemorrhagic *Escherichia coli* (EHEC) is a common pathotype of *E. coli* that causes numerous outbreaks of foodborne illnesses. EHEC is a zoonotic pathogen that is transmitted from animals to humans. Ruminants, particularly cattle, are considered important reservoirs for virulent EHEC strains. Humans can become infected with EHEC through the consumption of contaminated food and water or through direct contact with infected animals or humans. *E. coli* O157:H7 is one of the most commonly reported causes of foodborne illnesses in developed countries. The formation of attaching and effacing (A/E) lesions on the intestinal epithelium, combined with Shiga toxin production, is a hallmark of EHEC infection and can lead to lethal hemolytic–uremic syndrome (HUS). For the phage-dependent regulation of Shiga toxin production, antibiotic treatment is contraindicated, as it may exacerbate toxin production, limiting therapeutic options to supportive care. In response to this challenge and the growing threat of antibiotic resistance, phytochemicals have emerged as promising antivirulence agents. These plant-derived compounds target bacterial virulence mechanisms without promoting resistance. Therefore, the aim of this study is to summarize the recent knowledge on the use of phytochemicals targeting EHEC. We focused on the molecular basis of their action, targeting the principal virulence determinants of EHEC.

## 1. Introduction

Most known *Escherichia coli* strains are a part of the natural microbiome of human intestinal flora. Pathogenic *E. coli* strains are broadly classified based on their virulence factors and disease characteristics. The two major distinctions of pathogenic *E. coli* strains are non-diarrheagenic and diarrheagenic [1]. Pathogenic *E. coli* strains unrelated to gastroenteritis are mainly referred to as extraintestinal pathogenic *E. coli* (ExPEC), which are the etiological agent of infection outside the host’s gastrointestinal tract. ExPEC members comprise, among others, uropathogenic *E. coli* (UPEC), sepsis-associated *E. coli* (SEPEC), and neonatal meningitis-associated *E. coli* (NMEC) [1]. Diarrheagenic *E. coli* strains are associated with severe gastroenteritis and include the Shiga toxin-producing *E. coli* (STEC) and enterohemorrhagic *E. coli* (EHEC) groups that are the focus of this review. *E. coli* O157: H7, the most prominent EHEC member, has an extremely low infectious dose, estimated at 10 to 100 cells. Infections typically affect the elderly and children due to insufficiency of the immune system. However, the German outbreak in 2011 showed that age is not a limiting factor in the development of EHEC infections, as many of the several cases involved young and previously healthy persons [2]. Gastrointestinal symptoms usually manifest within 3–4 days post infection. The most common complication is hemorrhagic colitis, which is characterized by abdominal cramps and bloody diarrhea. STEC strains are estimated to cause more than 265,000 cases annually only in the United States, making them among the world’s greatest microbial threats. More serious sequela of EHEC infection is hemolytic-uremic syndrome (HUS), leading to hemolytic anemia, acute renal failure and thrombocytopenia, and subsequent kidney disfunction, resulting in the patient’s death or the necessity of kidney transplants [3].

The EHEC virulence factor that distinguishes HUS-inducing strains from other *E. coli* pathovars is Shiga toxin production. There are two genetically distinct types of Shiga toxins, Stx1 and Stx2, each with several subtypes. Stx2a and Stx2c are commonly associated with HUS development in humans. Shiga toxin belongs to the AB_5_ toxin family, which binds specific glycosphingolipids on human cells, including glomerular and brain endothelial cells. These receptors include globotriaosylceramide (Gb3, CD77), and to a lesser extent, globotetraosylceramide (Gb4). Stx toxins cause cell death by inhibiting protein translation [4]. Manifestations of systemic intoxication include vascular dysfunction and thrombus formation, leading to HUS. Furthermore, the genes encoding Stx are located on the lambdoid prophage, making treatment difficult (Figure 1). Namely, the use of antibiotics, especially those from the quinolone family, can worsen the symptoms of infection by triggering a bacterial SOS response and subsequently excising the phage genetic material along with the transcription of phage-related genes [5]. In this context, virulence factors such as Shiga toxin will be released even if antibiotics effectively inhibit bacterial growth. Therefore, the treatment of Shiga toxigenic infections is currently limited to supportive therapies such as rehydration and dialysis. However, new therapeutic approaches are being developed to improve management of infection and disease by targeting virulence.

Nowadays, a resolute effort should be made to search for new antimicrobial agents that allow us to replace known antibiotics, as they are losing their effectiveness against multidrug-resistant pathogens. Plant-derived secondary metabolites provide a very diverse framework that has been used to design new drugs, including antimicrobials [6]. In nature, secondary metabolites contribute to the systemic and induced defense of plants against bacteria, viruses, and fungi. Several phytochemicals belonging to classes such as coumarins, flavonoids, terpenoids, and alkaloids show inhibitory properties against many microorganisms. However, there is still a gap in the knowledge to be filled concerning the mechanistic principles of their action against specific pathogens. Thus, the aim of this work was to summarize the current studies on the activity of phytochemicals against Shiga toxigenic *E. coli*.

## 2. EHEC Virulence Factors

### 2.1. Shiga Toxin Production

The genes responsible for encoding Shiga toxins (*stx* genes) do not originate from the bacterial chromosome. Instead, these genes are incorporated into the genetic material of lambdoid bacteriophages (viruses related to the λ family) that have the ability to lysogenize *E. coli* strains. These particular phages are referred to as Shiga toxin-converting bacteriophages [7]. The expression of *stx* genes is directly dependent on the phage lytic cycle, which in turn is highly sensitive to environmental stimuli. Generally, the expression of nearly all phage genes, except for the *cI* gene, is significantly suppressed when the phage exists in the prophage state, due to the action of the CI repressor [5,8]. This repressor inhibits the major early phage promoters, pL and pR, which are essential for the expression of genes involved in the lytic cycle. One such gene, Q, encodes an anti-termination protein necessary for efficient transcription from the pR′ promoter, responsible for expression of so-called late-phage genes [8]. Transcripts originating from the pR′ promoter direct the synthesis of proteins involved in host cell lysis, along with structural proteins needed for capsid assembly. In Shiga toxin-converting bacteriophages, the *stx* genes are located between pR′ and the lysis genes [7,8]. Therefore, in lysogenized bacteria, the expression of *stx* genes is repressed, and the production of Shiga toxins can only occur following prophage induction [9,10]. Moreover, after prophage induction, the efficient expression of *stx* genes relies on successful phage DNA replication, leading to the amplification of these genes within the host cell [7].

### 2.2. LEE (Locus of Enterocyte Effacement)

In addition to the production of Stx toxin, the pathogenesis of EHEC is also characterized by the bacterium’s ability to colonize the human intestine [11,12]. After entering the gastrointestinal tract, EHEC migrates to the distal ileum and colon, where it colonizes these organs by forming strong attachments to the surface of epithelial cells. At sites of bacterial attachment to colonocytes, EHEC induces the formation of F-actin-rich pedestals directly beneath the attached bacteria [11]. Genes encoded on a pathogenicity island (PAI) within the locus of enterocyte effacement (LEE) are responsible for this characteristic EHEC phenotype. The LEE region contains 41 open reading frames organized into five polycistronic operons, LEE1 to LEE5. LEE expression is tightly regulated and encodes the type 3 secretion system (T3SS). The T3SS acts like a molecular syringe, delivering bacterial effector proteins directly into the cytoplasm of host cells [13]. The presence of LEE had been demonstrated also for other enteropathogenic *E. coli*, *Citrobacter rodentium*, and *E. albertii* [14].

When EHEC makes close contact with host epithelial cells, it employs the T3SS to inject dozens of effector proteins into the host cell’s cytoplasm. One key effector is the translocated intimin receptor (Tir), which integrates into the host plasma membrane and serves as a receptor for intimin, a bacterial outer membrane protein. The interaction between intimin and Tir facilitates bacterial attachment and triggers actin polymerization, resulting in the formation of characteristic attachment and effacement (A/E) lesions. The expression of LEE is controlled by the master regulator Ler, encoded by the LEE1 operon, and is further modulated by multiple transcription factors in response to environmental cues. Recent reviews have extensively covered the regulation of EHEC virulence factors [13,15,16].

### 2.3. Biofilm

Although biofilm formation is not a primary virulence feature of EHEC, its multicellular structure helps bacteria survive under unfavorable conditions [17]. The ability of *E. coli* O157:H7 to attach to abiotic and biotic surfaces through specific virulence factors—such as adhesins, flagella, type I fimbriae, curli fibers, and the enterocyte effacement locus—is a complex and critical step in infection initiation and biofilm formation [18]. This process is influenced by various factors, including the hydrophobicity and charge of the cell surface, as well as the amount and distribution of extracellular polymeric substances. Flagella enable bacterial motility and help EHEC overcome repulsive electrostatic forces between surfaces and cells, facilitating adhesion and biofilm formation on both abiotic and biotic surfaces [19]. Curli are thin, coiled fibers of varying lengths that self-assemble outside the bacterial cell and form an amorphous matrix essential for colonization and biofilm development. During attachment, the genes responsible for curli production are upregulated, enhancing surface association and providing protection against unfavorable environmental conditions. The production of curli polymers is environmentally regulated and depends on RpoS (sigma S subunit of RNA polymerase), meaning that *csgA* (the structural subunit of curli) transcription is controlled by environmental factors such as low temperature, osmolarity, and stationary phase growth conditions.

Adhesins, outer membrane proteins (such as intimin), and long polar fimbriae (LPF) play important roles in the colonization of human intestinal epithelial cells [16]. The bacterium has two chromosomal operons encoding LPF, which contribute to adherence in vitro and colonization in vivo. It also forms exopolysaccharides (EPSs) that act as a physical barrier to protect cells from environmental stressors such as temperature changes, nutrient limitations, pH changes, and biocide exposure. EPS is also involved in cell adhesion and biofilm formation, acting as a conditioning layer or adhesive on abiotic surfaces such as stainless steel.

The formation of biofilms by EHEC bacteria is regulated by quorum sensing (QS). QS is a system that globally controls the expression of genes involved in virulence factor secretion and biofilm maturation [20]. Bacterial cells secrete signaling molecules (autoinducers) into their surrounding environment. Once a critical density has been attained, the cells initiate the process of biofilms forming and maturing. The chemotactic attraction of EHEC by AI-2, a key autoinducer, is mediated by two receptors, LsrB and LsrR, which regulate motility, cell attachment, and biofilm formation. *E. coli* can interfere with the AI-2-regulated behavior of other bacterial species in an Lsr-dependent manner. The signaling properties of AI-2 enhance the adaptability of both Gram-positive and Gram-negative bacteria to various environmental conditions, promoting adhesion and biofilm formation, which in turn contributes to pathogenicity. AI-2, known as a “universal autoinducer”, regulates bacterial physiology and impacts virulence factor production and biofilm formation in certain bacteria. Another signaling molecule associated with *E. coli* O157:H7 is autoinducer-3 (AI-3), which requires tyrosine for synthesis and depends on LuxS to activate the transcription of pathogenicity genes located within the LEE.

## 3. Phytochemicals with Antimicrobial Activity Against EHEC

The treatment of EHEC infections poses unique challenges, as antibiotics can exacerbate disease severity by inducing Shiga toxin production machinery, as was mentioned above. This highlights the need for alternative therapeutic strategies that target bacterial pathogens through mechanisms distinct from existing antibiotics to avoid the risk of worsening a patient’s condition due to the treatment [21]. Antivirulence strategies represent an innovative alternative by targeting specific bacterial virulence mechanisms, such as toxin production, adhesion, or biofilm formation, without directly affecting bacterial survival [22,23,24]. This approach minimizes the risk of selective pressure that drives the emergence of antibiotic resistance and reduces the potential disruption of the host microbiome. Unlike conventional antibiotics, antivirulence agents are designed to disarm the pathogen, thereby mitigating its ability to cause disease while allowing the host immune system to overcome infection. This strategy is particularly critical in the face of the growing ineffectiveness of broad-spectrum antibiotics. Plant-derived phytochemicals, with their diverse structures and bioactive properties, represent a particularly promising class of antivirulence agents, which warrants further exploration of their mechanisms and therapeutic potential not only in the context of EHEC infections. In the following chapters, we focus on describing the anti-virulent properties of selected phytochemicals.

### 3.1. Isothiocyanates

Isothiocyanates (ITCs) are the group of compounds containing the monovalent anion [N=C=S]^−^, with the common formula of R–N=C=S; chemically, they are salts or esters of isocyanic acid. These secondary plant metabolites are considered stress-response plant chemicals. They are abundant in plants from the *Brassicaceae* family, particularly in the *Brassica* genus. Their precursors, glucosinolates, are hydrolyzed by the enzyme β-thioglucosidase (myrosinase), which is released upon plant tissue injury as a protection against the source of damage [25]. Another source of myrosinase, especially important for dietary ITCs, is the intestine microbiome; therefore, cooked vegetables, devoid of active myrosinase, can still be a source of ITCs. The popularity of *Brassicaceae* plants in the human diet in all regions of culture and climate (e.g., broccoli, cabbage, horseradish, kale, cauliflower, turnip, watercress, mustard, bok choy, and many others) has made ITCs a part of daily phytochemical intake and promoted the widespread knowledge of their health benefits. Each of the various glucosinolates present in plants leads to the formation of a different ITC upon hydrolysis, providing a variety of these bioactive compounds with particular characteristics (Figure 2). The increasing attention on this group of compounds is justified by their broad health-promoting features. The most commonly known ones are anti-inflammatory, anti-cancer, and antioxidant activities, as well as cancer prevention effects [26]. Especially, the chemopreventive effect of several ITCs, such as sulforaphane, has been extensively studied and attributed to detoxification of the cells (of, e.g., carcinogens) via the glutathione *S*-transferases (GSTs) facilitated by the ITCs; therefore, several widely promoted health campaigns have pointed to the beneficial effect of increasing consumption of vegetables such as broccoli or cabbage. Equally interesting is anti-inflammatory effect of ITCs, where the compounds exert their effects by modulating cytokine response, affecting toll-like receptors and transcription regulators on multiple levels [27]. Thus, as the currently increasing number of illnesses have been described as related to the general inflammatory response, such a role of widely available phytochemicals is invaluable. Therefore, extensive studies have been conducted to identify new compounds for therapeutical use among this group. Such a search is additionally facilitated by the reports from folk medicine of the beneficial effects of, e.g., mustard seed, horseradish, broccoli, and many others still included in dietary advice. Moreover, the safety of ITCs as a part of the human daily diet as well as in direct therapy has been assessed, along with the therapeutical doses used in, e.g., the inhibition of cancer cell invasion and angiogenesis, sensitizing cancer cells to chemotherapy, or directing cancer cells to cell cycle arrest, autophagy, and apoptosis. Clinical trials conducted recently have demonstrated the potential of ITCs to protect from or treat serious diseases such as breast, prostate, or lung cancer; inflammatory illnesses; diabetes; and heart disorders [28]. The antimicrobial effects of ITCs have recently attracted increased attention due to the dramatic decline in the introduction of new agents and the limited effectiveness of already existing ones. The effectiveness of ITCs against bacteria have been demonstrated for various species in vitro; in addition, a very limited level of microbial resistance to these compounds has been reported [29]. Moreover, clinical trials have documented the effect of ITCs in the treatment of urinary tract and respiratory infections [27], as separate compounds or in a combination of ITCs [30]. Various ITCs exhibit varying antibacterial effects, with phenethyl isothiocyanate being one of the most potent and studied against, e.g., human oral pathogens or bacteria from human guts [31,32]. EHEC strains as the target of ITCs have also been explored, aiming to establish alternative or at least supporting therapy to the risky usage of classical antibiotics. The approaches of employing ITCs against EHEC is described in the following paragraphs.

#### 3.1.1. Allyl Isothiocyanate

Allyl isothiocyanate (AITC), derived from glucosinolate sinigrin, is responsible for the specific pungent flavor and odor of, e.g., horseradish and mustard. Its antibacterial effect against foodborne pathogenic bacteria has been reported [33], prompting researchers to study the potential use of this ITC in food preservation. It was based on and supported by the traditional use of mustard seeds in long-term preservation of food products (mainly vegetables). AITC use in food preservation has been approved in Japan for over 20 years, providing it comes from a natural source [29]. More detailed studies focused on the specific antimicrobial applications of AITC. As EHEC strains have been often found in processed meat products, especially beef, which is also consumed uncooked or undercooked (e.g., beef tartare or widely eaten hamburgers), special caution has to be taken to prevent spreading the contamination during processing and storage. Other sources of EHEC contamination are vegetables, particularly leafy ones, used in ready-to-eat salads and sprouts. Actually, in the recent European outbreak of EHEC infection in 2011, which caused an unusual number of complication incidents, the etiological factor was identified as the EHEC O104:H4 strain, present in contaminated sprouted seeds [34]. The use of AITC as an agent preventing bacterial spread in packed food products such as ground beef has been systematically studied, providing satisfactory results of reducing the number of pre-inoculated *E. coli* O157:H7 by at least 3.0 logs with moderate stability of AITC after days of incubation in refrigerated or frozen products [35]. The use of microencapsulated AITC facilitated the handling of AITC but reduced its effectiveness in eliminating pathogenic strains. Interestingly, the number of microbiome-representative bacteria was not significantly altered by AITC [36]. Allyl isothiocyanate has been used also for the protection of vegetable products. For example, AITC has been showed to be an effective agent in eliminating EHEC bacteria from alfalfa seeds, and was then proposed to be employed as a disinfectant instead of as chlorine treatment [37]. The extended delivery of AITC to stored food was achieved by using encapsulation in calcium alginate beads, whose effectiveness was tested on spinach leaves [38]. Another potential application of AITC as a preventive countermeasure against EHEC takes advantage of the combined effect of several natural compounds. For example, a mix of garlic essential oil, AITC, and nisin was employed in the preservation of fresh meat products [39]. The higher effectiveness in microbial control of the mixed compounds (AITC, cinnamaldehyde, and carvacrol) in the vapor phase was shown for EHEC on lettuce and spinach leaves [40].

The proposed mechanism of AITC antibacterial effect in EHEC eradication was discussed to involve metabolic disturbance by inhibition of thioredoxin reductase (necessary in ribonucleotides synthesis) and acetate kinase (used in general energy metabolism) [41]. However, it has also been suggested that AITC affects cell membrane integrity, resulting in leakage of cell content and subsequent cell death [42]. AITC is shown to be active in the range of a minimum inhibitory concentration (MIC) from 100 to 400 µg/mL against *E. coli* O157:H7. The insight into the mechanism of AITC action was provided by the finding that *E. coli* mutants devoid of the BaeSR two-component system exhibit higher resistance to AITC. This system is responsible for controlling the expression of drug-exporting genes and sensing the environmental stress [43].

#### 3.1.2. Phenethyl Isothiocyanate

Phenethyl isothiocyanate is a product of gluconasturtiin hydrolysis, which is abundant in watercress. The antibacterial effect of this compound was shown for several bacterial pathogens causing foodborne diseases, including *Staphylococcus aureus*, *Pseudomonas aeruginosa*, *Listeria monocytogenes*, *E. coli*, and intestinal bacteria such as various clostridia [43,44]. Interestingly, the synergistic effect of PEITC and certain antibiotics was presented for ESBL *E coli* isolates [45]. The proposed PEITC mode of action suggests a direct effect on membrane integrity. This leads to cytoplasm leakage and K^+^ ion release [45]. The alterative mechanism was shown for EHEC strains by our group, explaining the bacterial growth inhibition by the global stress response directly and indirectly affecting cellular metabolism [46]. This mechanism is discussed in detail in the following chapters. Reported EHEC growth inhibitory PEITC concentrations ranged from 40 to 1000 µg/mL.

#### 3.1.3. Sulforaphane

Sulforaphane (SFN), and its precursor, glucoraphanin, are present in abundance in Brassicaceae plants common in the human diet, such as cabbage, broccoli, or Brussels sprouts. Its bio-active features are widely studied in terms of anticancer and chemopreventive roles, highlighting the SNF-mediated protection at all stages of carcinogenesis [47]. The antibacterial effect of SFN was reported for various respiratory pathogens such as *Streptococcus pneumoniae*, *Haemophilus influenzae*, and *S. pyogenes* [48]. The high effectiveness of SFN among other ITCs against enteropathogens, including EHEC (MIC ranging 350–700 µg/mL), prompted further studies encompassing its possible preventive and therapeutic use [31]. The antibacterial effect of SFN against Shiga toxin-producing *E. coli* strains have been studied by us, presenting novel mechanisms of ITC action [49] (see below).

#### 3.1.4. The Induction of the Stringent Response as a Universal Mechanism of ITC Antimicrobial Effect

The controversies and discrepancies among the proposed hypotheses about possible mechanisms of ITC effects and the quite limited number of reports elucidating these mechanisms led to more extensive investigation of this phenomenon—in particular, because of the necessity to fully understand the processes leading to bacterial growth and virulence inhibition before the compounds could be used in therapy. The observation that all tested ITCs varying in side residues (aromatic, aliphatic with various chain length) inhibited the nucleotide synthesis (both DNA and RNA), and in particular, that the production of RNA was dramatically downregulated just after the addition of ITC, prompted the hypothesis that the global regulatory mechanism of the stress response in bacterial cells can be a target of ITCs. The stringent response of the bacterial mechanism to deal with stress and various energy and macromolecules deprivation leads to the inhibition of stable RNA synthesis abruptly after the onset of stress conditions. The alarmones of this stress—unusual guanosine nucleotides, (p)ppGpp—are synthesized by widely conserved enzymes Rel and Spo, or bifunctional RSH [50]. The accumulation of (p)ppGpp was observed in *E. coli* cells treated with ITCs, namely, sulforaphane (SFN), allyl isothiocyanate (AITC), benzyl isothiocyanate (BITC), phenethyl isothiocyanate (PEITC), phenyl isothiocyanate (PITC), and isopropyl isothiocyanate (IPRITC) [46,51]. Growth of both wild-type *E. coli* and O157:H7 clinical isolates was inhibited by all tested ITCs, with the lowest MIC for PEITC and BITC. The bacterial growth inhibition was clearly attributed to (p)ppGpp directing the cellular metabolism to energy-saving mode. Interestingly, under this experimental setup, ITCs have not caused membrane disruption, which contradicts previous reports, nor did it increase reactive oxygen species formation. The latter finding is important in the light of the risk of DNA damage resulting in prophage induction and Shiga toxin synthesis. The most important, though, was finding that ITCs, while inhibiting bacterial growth, did not prompt *stx* gene expression [46,51], which was due to the efficient inhibition of prophage induction, with the main target being the transcription from early phage promoters such as pR. The impairment of phage lytic development under ITC treatment was observed even in the presence of the DNA damaging agent mitomycin C. As a result, the bacterial lysates treated with ITCs decreased significantly their virulence against HeLa and Vero cells. All above-mentioned effects were observed only when (p)ppGpp accumulated, and the effects were absent in mutants devoid of the functional RelA enzyme. RelA is responsive to amino acid starvation; thus, the hypothesis was proposed that ITCs induce a stringent response by blocking protein synthesis either by direct interaction with amino acids (which was previously shown for AITC) or by the impairment of tRNA aminoacyl transferase or synthetase. This is supported by the finding that an excess of certain amino acids reversed the antibacterial effect of ITCs. Further work on various aliphatic ITCs, such as SFN, sulforaphene (SFE), iberverin (IBR), iberin (IBN), alyssin (ALN), erucin (ERU), erysolin (ERY), and cheirolin (CHE), support this hypothesis, as all tested compounds induced the stringent response, causing bacterial growth and prophage induction inhibition in various tested clinical isolates of *E. coli* O157:H7. Interestingly, a synergistic effect was observed for some ITC combinations [52].

The finding of the apparently common mechanism of ITC action on EHEC strains provides very interesting input in the debate on the treatment of EHEC infections. First, ITCs can be considered an option of direct therapeutical agents, especially due to previous extensive studies on their safe usage. Then, the specific mechanism of ITC action can be employed in combined therapy with ITCs and antibiotics to reduce or fully exclude the risk of phage lytic development induction and subsequent Shiga toxin synthesis.

### 3.2. Phenolic Compounds

Common phenolic compounds include (i) polyphenols, such as lignin and tannins; (ii) oligophenols, such as flavonoids, stilbenes, and coumarins; and (iii) monophenols (simple phenolic acids), such as benzoic acid derivatives (hydroxybenzoic acids) and cinnamic acid derivatives (hydroxycinnamic acids). Phenols and polyphenols are a diverse group of chemical compounds characterized by the presence of at least one aromatic ring and hydroxyl groups (Figure 3). Based on the number and arrangement of carbon atoms, they are classified into 10 major classes [53,54]. This structural diversity translates into a broad spectrum of biological functions. To date, more than 8000 polyphenols have been identified, known not only for their potent antioxidant properties but also for their antimicrobial, antifungal, and antibacterial activities [55,56]. In addition to their direct impact on metabolic pathways, polyphenols also exert anti-inflammatory effects by downregulating production of pro-inflammatory cytokines and modulating immune responses. Their antioxidant potential extends to protecting cells from oxidative damage, which is linked to a reduction in the risk of neurodegenerative disorders such as Alzheimer’s and Parkinson’s disease and the prevention of cardiovascular diseases and cancers [57]. Recent studies have also explored their role in improving gut health, as polyphenols can influence the composition of the intestinal microbiota, promoting beneficial bacteria and suppressing pathogenic strains [58]. Furthermore, they have the ability to modulate various signaling pathways, including the regulation of cytochrome P450 enzyme activity, which influences drug metabolism [59,60]. For example, epigallocatechin gallate (EGCg) inhibits the activity of cytochrome P450 enzymes, while bergamottin, a natural compound found in grapefruit, inhibits the CYP2C19 isoenzyme, affecting detoxification mechanisms. A significant part of the research explores the impact of polyphenols on pathogenic *E. coli*, particularly enterohemorrhagic strains (as shown in Table 1).

#### 3.2.1. Flavonoids

Among the polyphenols, flavonoids have been particularly studied for their biofilm-inhibiting effects. Jin-Hyung Lee and colleagues tested nine flavonoids for their impact on *E. coli* O157:H7 biofilm formation. The study also included antioxidant compounds such as vitamins C and E, which did not affect biofilm formation, suggesting that the biofilm-inhibiting properties of flavonoids were not related to their antioxidant activity. Phloretin emerged as the most potent flavonoid, reducing EHEC biofilm formation by 89% and 93% at concentrations of 25 and 50 μg/mL, respectively. Interestingly, phloretin did not inhibit biofilm formation in four strains of *E. coli* K-12 and non-pathogenic *E. coli* strains; in fact, biofilm formation was enhanced. These findings were supported by qRT-PCR analysis, which showed that genes repressed in the biofilm cells of *E. coli* O157:H7 (*hlyE*, *lsrA*, *lsrB*, *csgA*, and *csgB*) were strongly induced in the biofilm cells of *E. coli* K-12. DNA microarray data also indicated that phloretin inhibited the expression of Shiga toxin 2 (Stx2) genes. Importantly, their results confirmed that phloretin exhibited anti-inflammatory properties in both in vitro and in vivo models of colitis.

Vikrama et al. [61] studied several other flavonoids, including naringenin, kaempferol, quercetin, and apigenin. These compounds inhibited the activity of the AI-2 signaling molecule in *Vibrio harveyi*, indicating that they are intercellular signaling inhibitors (autoinducers), which are critical in biofilm formation. These flavonoids also reduced biofilm formation by *V. harveyi* and *E. coli* O157:H7.

As demonstrated above, the beneficial effects of polyphenols on EHEC do not necessarily rely solely on their bactericidal action. Another potential mechanism exhibited by polyphenols is their ability to bind directly to Shiga toxin, thereby inactivating it. Dong et al. [63] demonstrated that baicalin (BAI), a flavonoid isolated from *Scutellaria macrantha* Fisch., induces oligomer formation in the toxin through direct binding. In vitro studies using HeLa cells showed that when BAI was either mixed with rStx2 prior to cell addition or incubated with HeLa cells treated with 50 pg/mL rStx2, it significantly protected the cells at concentrations of 4.5 μM, with maximum protection observed at 36 μM. Importantly, in a mouse model, BAI administration (100 mg/kg dose) every 6 h for a total of 23 doses following rStx2 exposure provided approximately 70% protection by day 6. Mice that did not receive BAI treatment showed significant kidney damage, including tubular swelling and hyaline casts, while kidneys in BAI-treated mice appeared similar to those of control animals not exposed to the toxin. Further efficacy of BAI against *E. coli* O157:H7 infections was demonstrated in both tissue culture and mouse models [62]. In tissue culture, BAI addition protected HeLa cells from Stx-induced damage, with significant protection at 9 μM and maximum protection at 72 μM. In a mouse model, BAI co-inoculation with *E. coli* O157:H7 and mitomycin C (which induced 100% mortality within 8 days in untreated, infected animals) reduced mortality to 20% on day 8, thus providing approximately 80% protection. Histopathological examination revealed that untreated, infected mice showed kidney damage marked by swelling, discoloration, and necrosis, while the BAI-treated group showed significantly reduced damage. Another study by Vinh et al. investigated baicalein, a structural analogue of BAI [64]. When baicalein was pre-incubated with Shiga toxin and then added to Vero cell cultures, Vero cell viability decreased in response to increasing concentrations of Stx1 and Stx2. However, baicalein pre-treatment significantly reduced Stx1 and Stx2 cytotoxicity, protecting Vero cells by binding to the surface of their cytoplasmic membranes and altering membrane function. At a concentration of 0.13 mmol/L, pre-incubation with baicalein decreased Vero cell sensitivity to Stx1 and Stx2 by approximately 6- and 8-fold, respectively. Molecular docking analysis suggested that baicalein forms a complex with both Stx1B and Stx2B pentamers. However, in cellular models, Stx binding to the Vero cell surface was preferred over baicalein binding to StxB pentamers, which aligns with the observed results. Interestingly, gene expression analysis revealed that baicalein slightly increased the transcription of *stx1* but not *stx2*, compared to a negative control [64].

Additionally, baicalin and baicalein exhibit a range of pharmacological activities beyond their effects on Shiga toxin. These compounds help reducing inflammation during bacterial infections by scavenging reactive oxygen species (ROS) [101]. Baicalin also synergizes with various antibiotics by disrupting bacterial tolerance mechanisms, enhancing bactericidal effects through membrane disruption, efflux pump inhibition, and biofilm formation reduction [102]. Interestingly, baicalin shows antimicrobial activity against *Helicobacter pylori* but does not affect probiotic strains, and it even promotes biomass growth in *Lactobacillus casei* [103].

#### 3.2.2. Resveratrol

Resveratrol, a polyphenolic stilbene compound found in foods such as red grapes and berries, is largely known for its health benefits as an antioxidant [104,105]. However, recent studies have also highlighted its significant antimicrobial activity against EHEC [67]. Lee et al. demonstrated antibiofilm properties of resveratrol toward EHEC [91]. Transcriptional analyses showed that the extract of *Carex dimorpholepis* Steud. repressed curli genes, various motility genes, and AI-2 quorum-sensing genes, which was corroborated by a reduction in the production of fimbria, motility, and biofilm by EHEC. Trans-resveratrol at 10 μg/mL in the extract of *C. dimorpholepis* was found to be a potent anti-biofilm compound against EHEC. Such biofilm inhibition was not limited to resveratrol alone; its oligomeric derivatives, including dimers and tetramers, showed even stronger anti-biofilm activity [90,92]. Furthermore, resveratrol-enriched extract was found to significantly decrease the adhesion of EHEC to human epithelial cells without compromising host cell viability. This dual action—reducing pathogen adhesion and maintaining host cell health—highlights resveratrol’s potential as a safe and effective antimicrobial agent in preventing EHEC infections [91]. These findings underscore the potential of resveratrol and its derivatives as natural antimicrobials in both clinical and food safety applications, providing a multifaceted approach to combating EHEC.

#### 3.2.3. Cinnamaldehyde

Cinnamaldehyde (*t*-CA), also known as 3-phenyl-2-propenal, is an unsaturated aldehyde belonging to the phenylpropanoid class. The history of use of cinnamon (with *t*-CA as its main content) reaches ancient times and included various areas and continents; in its use, the antimicrobial and preservation-promoting features of cinnamon have been utilized. It involved the application of cinnamon as a spice, in religious ceremonies, in fragrances, as a digestive help, and as an appetite stimulant. Currently, *t*-CA is widely used mostly as a spice and thus is considered a safe daily diet component. The beneficial health role of *t*-CA includes, but is not limited, to its antioxidant, anti-inflammatory, and anti-diabetic action [106]. Research indicates that *t*-CA affects membrane potential and permeability, leading to protein oxidation, cytoplasmic leakage, and acidification, thereby reducing the toxicity of EHEC strains [86]. Additional studies have shown that sublethal concentrations of *t*-CA modulate the expression of virulence genes in *E. coli* O157:H7 and reduce bacterial motility (e.g., *fliA* and *motA* genes) [85]. An in-depth investigation into the mechanism of cinnamaldehyde’s action against EHEC strains revealed that reduced virulence is associated with the suppression of Stx prophage induction and Shiga toxin expression. This is mediated through the stringent response, and its alarmone molecules (p)ppGpp, which is activated by *t*-CA [106]. The study from our lab shows that the activity of *t*-CA led to amino acid depletion and thereby cell starvation by triggering acidic stress conditions. We showed in the study that *t*-CA activity led to impairment in metabolic processes as a tricarboxylic acid (TCA) cycle and energy-consuming processes like flagellar assembly. We assumed the observed effects were consequences of global gene expression regulation mediated by (p)ppGpp. Importantly, alarmones of the stringent response are the inhibition factors of the major lytic promoter (pR) of lambdoid phages [106]. Mechanistically, *t*-CA activity disrupts intracellular pH homeostasis, prompting the bacterial cell to utilize amino acids such as L-lysine and L-serine as part of its defense mechanisms. Consequently, this leads to depletion of these key amino acids and activation of RelA-dependent alarmone production. This cascade of events not only impacts bacterial metabolism but also compromises the ability of the pathogen to sustain its virulence. Numerous studies have investigated *t*-CA applications in the food industry, where it is particularly valued for its antimicrobial properties. It has demonstrated potential efficacy against various strains of *S. aureus*, *Salmonella* spp., *Bacillus* spp., and *E. coli*, including EHEC strains [83,84]. For example, a study by Manu et al. showed that cinnamaldehyde (1.5 μL/mL) at 4 °C completely inactivated *E. coli* O157:H7 inoculated in carrot juice within 8 h [82]. Similarly, Baskaran et al. reported that at 4 °C, cinnamaldehyde concentrations of 0.125% and 0.075% *v*/*v* reduced bacterial counts in apple juice and cider to undetectable levels by days 3 and 5 of the experiment, respectively [81]. EHEC strains are frequently detected on vegetables; however, treatment of leafy greens with *t*-CA has been shown to effectively reduce these pathogens in an environmentally friendly manner [81].

#### 3.2.4. Other Phenolics

While the antimicrobial activity of many natural phenolics has been established, studies aimed at implementing such compounds against EHEC are especially valuable when they extend beyond in vitro tests and use animal models, as the effects of the compounds in such systems may differ from expected outcomes. For example, Kang-Mu Lee and colleagues [107] confirmed the inhibitory effect of epigallocatechin gallate (EGCg) from green tea on the virulence of EHEC strains using the *Caenorhabditis elegans* model. The nematodes were exposed to the pathogen with and without EGCg. After 5 days, survival rates of *C. elegans* were 47.1% without EGCg and 76% with EGCg. Furthermore, the researchers observed a reduction in the transcription of several virulence genes *(eae*, *escN*, *espA*, *sepZ*, and *tir*), regulated by quorum sensing at a concentration of 25 μg/mL—lower than the MIC (539 ± 22 μg/mL) they determined. The transcription levels of these genes decreased by 72.1%, 87.3%, 79.5%, 78.8%, and 90.8%, respectively.

The same model was used to investigate the effects of other compounds, such as coumarin, esculetin, and umbelliferone [80]. Esculetin demonstrated a dose-dependent effect on the survival of infected *C. elegans*, which corresponded to the repression of the *stx2* gene observed in qRT-PCR experiments. However, despite showing antibacterial properties against EHEC, neither coumarin nor umbelliferone affected the survival of *C. elegans* or the expression of Shiga toxin genes. Interestingly, both coumarin and umbelliferone, at concentrations of up to 50 μg/mL, exhibited high anti-biofilm activity without inhibiting planktonic cell growth at the same concentration. This indicates that the biofilm reduction is associated with an anti-biofilm mechanism rather than bacterial growth inhibition.

These studies confirm that, due to the virulence mechanisms of EHEC strains, simply determining the antibacterial properties of a compound is not sufficient. It is essential to thoroughly investigate the mechanisms of their action. Overall, studies on the impact of polyphenolic compounds on EHEC bacteria have often focused on their anti-biofilm properties. While *E. coli* O157:H7 is non-invasive and poorly forms biofilms, rare O157 strains that produce curli exhibit strong biofilm formation at 37 °C and possess the ability to invade epithelial cells. This correlates with a single-base-pair transversion (A to T) in the curli promoter [18]. Additionally, biofilm formation can be crucial in terms of its development on food products, which are known to be carriers of EHEC and may serve as a source of infection [108,109].

Other studies [110] focused on the binding of compounds to the toxin, investigating galloylated catechins (GCg) and epigallocatechin gallate (EGCg). Both compounds inhibited Stx1 cytotoxicity in Vero cells after pre-incubation at a concentration of 100 mg/L but did not show the same activity against Stx2. In silico studies indicated that EGCg formed a more stable structure with the Stx1B pentamer than with the Stx2B pentamer.

The use of polyphenols as natural preservatives in food processing is gaining increasing attention. Numerous studies highlight their antibacterial properties, demonstrating their ability to effectively combat both Gram-positive and Gram-negative bacteria. Additionally, some polyphenols exhibit synergistic effects when used in combination with traditional food preservatives or processing methods [53]. In a study by Wu et al., it was found that cranberry concentrate, containing high levels of polyphenols, at concentrations of 75 μL/mL and 100 μL/mL reduced *E. coli* O157:H7 by 0.8 log and 3.0 log CFU/mL, respectively, compared to the control after 24 h [111]. Additionally, cranberry concentrate (at 2.5%, 5%, and 7.5% *w*/*w*) reduced the number of *E. coli* O157:H7 inoculated into ground beef by 0.4 log, 0.7 log, and 2.4 log CFU/g, respectively, compared to the control on day 5 [112]. To further investigate the direct effect of polyphenols on EHEC, another study fractionated cranberry extract to isolate its polyphenolic components and evaluated their impact independently. Results showed that the polyphenol fractions reduced the bacterial count to below detectable levels at concentrations of 5.40 g/L and 2.70 g/L compared to the control. *E. coli* O157:H7 cells treated with polyphenols derived from *Vaccinium macrocarpon* Aiton (cranberry) exhibited localized damage to the outer membrane, cytoplasmic leakage, and irregular cell shape [113].

### 3.3. Terpenes and Terpenoids

Terpenes and terpenoids are naturally occurring compounds found in various plants, including fruits, vegetables, and herbs. They contribute to the characteristic flavors and aromas of these plants and often have therapeutic properties [114]. Some terpenes and terpenoids have been studied for their potential antibacterial properties, including their effects against pathogenic bacteria such as *E. coli*, and specifically Shiga toxigenic strains [40,115]. Terpenes are hydrocarbons derived from the combination of isoprene units (C_5_) (Figure 4), while terpenoids are modified terpenes that have undergone oxidation or other chemical alterations. Both terpenes and terpenoids can exhibit antimicrobial activity through various mechanisms, including disrupting bacterial cell membranes, interfering with cell wall synthesis, and inhibiting essential enzymes [114,115,116].

Several studies have investigated the antibacterial effects of terpenes and terpenoids against *E. coli*, including EHEC (as shown in Table 2).

#### 3.3.1. Carvacrol

Carvacrol is a natural monoterpenoid phenol. It is a constituent found in the essential oils (EOs) of several aromatic plants, with oregano (*Origanum vulgare* L.) and thyme (*Thymus vulgaris* L.) being notable sources. The essential oils extrackarted from these plants have been used for centuries in traditional medicine and culinary applications. The chemical structure of carvacrol consists of a phenol group (a hydroxyl group attached to an aromatic ring) and a monoterpene backbone. Its molecular formula is C_10_H_14_O, and it has the IUPAC name 2-methyl-5-(1-methylethyl)phenol [140]. Carvacrol has gained attention for its potential health benefits, particularly its antimicrobial properties. Studies have suggested that carvacrol exhibits antibacterial, antifungal, and antiviral activities. It may disrupt the cell membranes of microorganisms, interfere with their enzyme systems, and contribute to their overall antimicrobial effects [140,141].

Carvacrol’s efficacy against *E. coli* O157:H7 was investigated across various vegetables. The experiment by Landau and Shapira from 2012 involved exposing the bacteria to different concentrations of carvacrol in the vapor phase within a sealed container at temperatures of 0, 4, and 10 °C. On the intact surface of lettuce, even the lowest concentration of carvacrol vapor exhibited efficacy, deactivating over 4.0 log of *E. coli* O157:H7 within 4 days at 0 and 4 °C, and within 2 days at 10 °C. In contrast, on damaged tissue resulting from cutting, the highest concentration demonstrated a substantial reduction in the pathogen population by 4.0 log at 0 °C and 2.0 to 4.0 log at 4 °C over a 4-day period. At 10 °C, these concentrations also achieved a notable decrease in the pathogen population by 1 to 3.0 log within 2 days. Interestingly, the study revealed a more pronounced inactivation effect on lettuce compared to spinach leaves and on leaf surfaces compared to damaged areas in general. This highlights the impact of carvacrol concentration and temperature on different vegetable types and conditions. The findings suggest that utilizing antimicrobials in the vapor phase, specifically employing carvacrol, holds promise for enhancing the safety of refrigerated leafy greens when sold in sealed packaging. This approach could potentially contribute to mitigating the risk of *E. coli* contamination in such food products [129]. Another study by Mith et al. from 2015 investigated the impact of carvacrol on the expression of virulence-associated genes in *E. coli* O157:H7 (ATCC strain 35150), and the study revealed notable effects. Carvacrol exhibited effectiveness by firstly inhibiting the transcription of the *ler* gene, crucial for the upregulation of LEE2, LEE3, and LEE4 promoters, as well as the formation of attaching and effacing lesions. Secondly, it demonstrated a reduction in the expression of both Shiga toxin and *fliC* genes. Additionally, a discernible decrease in *luxS* gene transcription, associated with quorum sensing, was observed. These effects were found to be dose-dependent, highlighting a specific capability of carvacrol to downregulate the expression of virulence genes in EHEC O157:H7. These outcomes suggest that carvacrol holds potential to alleviate the adverse health consequences associated with virulence gene expression in EHEC O157:H7. Considering these findings, the application of carvacrol as a natural antibacterial additive in food or as an alternative to antibiotics emerges as a promising avenue for mitigating the impact of virulence in EHEC O157:H7 [120]. A study by Burt et al. (2005) examined carvacrol bacteriostatic and bactericidal properties with MICs of 1.2 mmol/L and additive in combination. Growth curves in the presence of nonlethal concentrations of carvacrol with the addition of agar (0.05%, *w*/*v*) or carrageenan (0.125%, *w*/*v*) as a stabilizer were produced by optical density measurement. The stabilizers agar and carrageenan significantly improved the effectiveness of carvacrol in broth, possibly because of a delay in the separation of the hydrophobic substrate from the aqueous phase of the medium. When carvacrol was dissolved in ethanol before being added to broth, stabilizers were not needed. Carvacrol and thymol, particularly when used in combination with a stabilizer or in an ethanol solution, may be effective in reducing the number of or preventing the growth of *E. coli* O157:H7 in liquid foods [121].

Another study focused on the determination of alteration in the membrane fatty acid profile as an adaptive mechanism of the cells in the presence of a sublethal concentration of an antimicrobial compound in response to a stress condition. Methanolic solutions of carvacrol were added to growth media of *E. coli* O157:H7 (ATCC 43888). The total percentage of unsaturated fatty acids was slightly increased [94]. Another study provides input on the mechanism of action of carvacrol. Namely, in the presence of 1 mM carvacrol during overnight incubation, bacteria produced significant amounts of heat shock protein 60 (HSP60) (GroEL) (*p* < 0.05) and significantly inhibited the synthesis of flagellin (*p* < 0.001), causing cells to be aflagellate and therefore nonmotile [142]. A study by Baskaran et al. investigated the effect of subinhibitory concentrations of carvacrol on production and virulence gene expression. Carvacrol reduced EHEC motility and attachment to human intestinal epithelial cells (*p* < 0.05) and decreased toxin synthesis by EHEC. The RT-PCR data revealed decreased expression of critical virulence genes in EHEC (*p* < 0.05). The results collectively suggest that carvacrol could be used to reduce EHEC virulence [143].

Carvacrol has undergone testing across various food products, including meat. The process of heating meat at high temperatures or for extended periods to eliminate foodborne pathogens can lead to the formation of potentially carcinogenic heterocyclic amines. To address this issue, ground beef was supplemented with 1% carvacrol, thoroughly mixed, and then inoculated with *E. coli* O157:H7. Subsequently, beef patties were prepared and compared to controls, resulting in a reduction in *E. coli* O157:H7 population of 2.5–5.0 log. The most significant reductions in three major amines, MeIQ (58%), MeIQx (72%), and PhIP (78%), were observed. These findings demonstrate that carvacrol effectively reduced both bacterial count and amine amount in a commonly consumed meat product [144]. Further research indicated that achieving a 4.0 log reduction in contaminated processed ground beef requires heating to an internal temperature of 60 °C for at least 30 min. The sensitivity to heat significantly increased (*p* < 0.05) with the addition of and/or increasing levels of carvacrol from 0.5 to 1.0%. These data can help in establishing critical control points for ground beef with lower heating times and temperatures [117].

Another study explored storage conditions and physicochemical parameters. Beef slices inoculated with *E. coli* O157:H7 were marinated in teriyaki sauce with or without 0.3–0.5% carvacrol. After 1, 3, and 7 days at 4 °C, the indigenous microflora population, color, lipid oxidation, marinade uptake, and pH of the marinated beef and leftover marinade samples were assessed. While teriyaki sauce alone did not reduce or inhibit bacteria, teriyaki sauce containing 0.5% carvacrol inactivated all inocula without recovery within 7 days (*p* < 0.05). Physicochemical parameters were not significantly affected (*p* < 0.05) [145]. In 2016, Moon and Rhee investigated the synergistic effect of soy sauce and carvacrol. Carvacrol (1 mM) eradicated all test bacteria within 1–5 min at 22 °C and within 10 min at 4 °C. The results suggest that carvacrol synergizes with other components present in soy sauce, possibly due to the combination of factors such as high salt concentration and low pH imparted by organic acids in soy sauce, along with the membrane-attacking properties of carvacrol [146] Another team studied the edible film of *Gelidium corneum* (Clemente Thuret) containing carvacrol as an antimicrobial and antioxidative agent. Increasing amounts of the antimicrobial agent enhanced its activity against *E. coli* O157:H7. Application of the film to ham packaging effectively inhibited microbial growth and lipid oxidation during storage [147].

Due to these properties, carvacrol has been investigated for its potential in food preservation, as well as in the development of natural antimicrobial agents. It is important to note that while carvacrol looks promising, further research is needed to fully understand its mechanisms of action and its applications in various fields. Additionally, the concentration and formulation of carvacrol can influence its efficacy and safety [140].

#### 3.3.2. Thymol

Thymol is a natural monoterpene phenol that is found in several plants, with thyme (*Thymus vulgaris* L.) being one of the most notable sources. It is widely recognized for its strong, aromatic odor and flavor, which makes it a popular ingredient in various culinary and medicinal preparations [148,149].

The impact of pH, temperature, water activity, sodium chloride levels, inoculum size, and the presence of competing microorganisms on thymol’s effectiveness against *E. coli* O157:H7 was investigated. It was observed that the susceptibility of *E. coli* O157:H7 to thymol increased as storage temperature, water activity, pH, and inoculum size decreased. Sodium chloride concentrations ranging from 0.5% to 2.5% and the presence of a microflora mixture did not significantly alter thymol antimicrobial effects on *E. coli* O157:H7 (*p* < 0.05). The MIC ranged between 500 and 1000 μg/mL depending on the EHEC strain [127]. Yuan and Yuk conducted a study investigating the impact of sublethal thymol exposure on virulence gene expression and virulence traits of *E. coli* O157:H7. Their findings indicated that bacteria exposed to sublethal concentrations of thymol during early stationary phase exhibited significantly reduced motility (which was reversible after stress removal), decreased biofilm formation, and diminished efflux pump activity. There was no observed induction of antibiotic resistance, and no significant alterations in adhesion and invasion capabilities on a human colon adenocarcinoma (Caco-2) cell line. qRT-PCR analysis demonstrated decreased expression of relevant virulence genes, including those associated with flagellar biosynthesis and function, biofilm formation regulation, multidrug efflux pumps, and components of the type III secretion system [85]. Similarly to carvacrol, thymol can effectively combat EHEC cells without triggering excessive production of toxins [143].

Essential oils pose a challenge in dispersing evenly in water due to their low solubility, leading to a heightened affinity for binding with food lipids and proteins. This diminished solubility reduces their antimicrobial effectiveness [95,150]. Researchers tested a nano-dispersed form encapsulated in whey protein isolate–maltodextrin conjugates against *E. coli* O157:H7 strains ATCC 43889 and 43894. The studies revealed that both nano-dispersed and free thymol exhibited an MIC of 300–500 ppm at pH 6.8 and a bacterial count reduction of 1.0 to 3.0 log CFU/mL after 48 h. At pH levels of 5.5 and 3.5, temperature variations did not significantly affect the antimicrobial activity of 500 mg/mL nano-dispersed thymol. The study highlighted the potent antimicrobial activity of transparent thymol nano-dispersions against various foodborne pathogens [126]. Another investigation explored nanoemulsions prepared using a mixture of ethyl lauroyl arginate (LAE) and lecithin. This combination resulted in stable translucent nanoemulsions of thymol with spherical droplets smaller than 100 nm, in contrast to the turbid emulsions formed using individual emulsifiers. Zeta-potential data suggested the formation of LAE–lecithin complexes, possibly through hydrophobic interaction. In 2% reduced-fat milk, nanoemulsions exhibited comparable antilisterial activities to free LAE in inhibiting *L. monocytogenes*. However, they were less effective against *E. coli* O157:H7 than free LAE, indicating a correlation with the availability of LAE, as observed in release kinetics. Therefore, while mixing LAE with lecithin improved the physical properties of EOC nanoemulsions, it did not enhance antimicrobial activities, especially against Gram-negative bacteria [95]. The nano form of essential oils provide a promising alternative with improved properties.

There are studies proving thymol’s efficiency in food products. Pathogenic *E. coli* strains commonly encountered in meat and poultry include intestinal pathogenic *E. coli* (EHEC, but called iPEC in this study) and extraintestinal types such as uropathogenic *E. coli* (UPEC). In Chien et al.’s study, researchers compared the resistance of iPEC (O157:H7) to UPEC in chicken meat using high-pressure processing (HPP) both with and without thymol essential oil as a sensitizer, following the hurdle concept. UPEC exhibited slightly higher resistance than *E. coli* O157:H7 at 450 and 500 MPa. A central composite experimental design was employed to assess the impact of pressure (300–400 MPa), thymol concentration (100–200 ppm), and pressure-holding time (10–20 min) on the inactivation of iPEC and UPEC strains in ground chicken. The hurdle approach aimed to reduce the intensity of high pressure and thymol levels applied to the food matrices, potentially minimizing damage to food quality post-treatment. The findings offer valuable insights into the survival of both O157:H7 and UPEC during HPP, whether in the presence or absence of thymol [151]. Osali et al. conducted a study to evaluate the antimicrobial properties of thymol (TH), an active essential oil, against *E. coli* O157:H7 in chicken tawook during storage at 4 and 10 °C. A marinade, comprising typical ingredients used in chicken tawook recipes, was prepared and combined with 1% and 2% *v*/*v* TH. This marinade, with or without essential oils (EOs), was applied to fresh chicken breast cubes inoculated with the foodborne pathogens. During storage at 10 °C, the marinade led to a reduction in *E. coli* O157:H7 levels on the chicken samples of approximately 3.3 log CFU/g on days 4 and 7 compared to untreated samples. At 4 °C, thymol contributed to a decrease in EHEC populations in chicken tawook of ≤2.4 log CFU/g compared to samples without it, where the reduction was ≤1.4 log CFU/g, regardless of storage duration. However, the addition of EOs did not significantly decrease *E. coli* O157:H7 populations in samples stored at 10 °C. Furthermore, increasing the concentration of EOs from 1% to 2% did not appear to have a significant impact on reducing the populations of the tested foodborne pathogens [152] The study from 2021 examined the impact of a yogurt-based marinade combined with active essential oils—thymol, carvacrol, and cinnamaldehyde—on *E. coli* O157:H7 in camel meat stored at 4 and 10 °C. At 4 °C, marinating meat did not significantly alter bacterial counts, but at 10 °C, a notable decrease occurred on days 4 and 7 of storage, ranging from 2.7 to 2.1 CFU/g compared to untreated CM. Incorporating EOs enhanced the reduction in *E. coli* O157:H7, with a more pronounced effect observed when the EO concentration increased from 1% to 2% [153].

Thymol antimicrobial properties have led to its widespread use in the food industry as a natural preservative and in healthcare settings for its disinfectant properties. It is commonly found in mouthwashes, topical antiseptics, and disinfectant solutions. Overall, thymol represents a promising natural compound with potent antibacterial properties, and ongoing research continues to explore its therapeutic potential in various applications [141,148,149].

#### 3.3.3. Menthol

Menthol is a naturally occurring compound found in the essential oil of mint plants, particularly peppermint (*Mentha piperita* L.) and spearmint (*M. spicata* L.). It is commonly used in various products such as cough drops, balms, and topical analgesics due to its cooling and soothing properties [154,155,156]. While menthol is primarily known for its sensory effects, there is limited evidence suggesting that it may possess certain antibacterial properties [154,157]. Some studies have explored the potential antimicrobial effects of menthol against various bacteria.

In a study from 2010, Landau and Shapira [129] discovered that preconditioning EHEC strains in escalating subinhibitory concentrations of menthol significantly enhances their resistance to menthol by 4- to 16-fold. Concurrently, observable morphological changes include the emergence of mucoid colonies and diminished biofilm production. Scanning electron microscopy (SEM) analysis reveals inhibited curli formation in cells adapted to menthol. In a laboratory *E. coli* strain, the expression of the *cpsB10* gene, responsible for colanic acid production, increases in response to SI menthol concentrations, while an *rcsC* mutant exhibits reduced expression, suggesting a partial involvement of the Rcs phosphorelay system in mediating the menthol signal. Adaptation to menthol also leads to decreased expression of the locus of enterocyte effacement-encoded regulator (Ler). This decrease, coupled with reduced curli and biofilm formation and increased mucoidity, implies a general reduction in bacterial virulence after menthol adaptation. Consequently, these findings proposed menthol as a potential candidate in the emerging alternative approach of targeting bacterial virulence factors for the development of novel anti-infective agents [129]. Menthol MIC has been determined to be 1.0 and a minimum bactericidal concentration (MBC) to be 2.0 μg/mL [119]. The antimicrobial activity of menthol is thought to be related to its ability to disrupt bacterial cell membranes and interfere with microbial growth. However, it is essential to note that the research on menthol antibacterial properties is not as extensive as that for other compounds, and more studies are needed to confirm and fully understand its mechanisms of action.

#### 3.3.4. Linalool

Linalool is a naturally occurring terpene alcohol that belongs to the class of compounds known as monoterpenes. It is commonly found in the essential oils of various plants, including lavender, basil, rosewood, and coriander. While linalool is primarily recognized for its aromatic properties, there is some evidence to suggest that it may possess certain antibacterial properties [158]. Several studies have investigated the potential antimicrobial effects of linalool against various bacteria.

In their work, Soković et al. examined several essential oils and its main components. Linalool (27.2%) and linalyl acetate (27.5%) are the most abundant components in *Lavandula angustifolia* Mill. oil (yield is 3% (*v*/*w*). Linalool is also the main component in *Ocimum basilicum* L. oil, with 69.3% (yield is 0.5% (*v*/*w*) [119]. In the disc-diffusion method, the inhibition zone for linalool was 12.0 mm, and the MIC against *E. coli* O157:H7 was determined to be 6.0 and the MBC 7.0 μg/mL [119]. The mechanism behind linalool’s antibacterial properties is not fully understood, but it is believed to involve disruption of bacterial cell membranes, interference with membrane permeability, and inhibition of bacterial enzymes. Additionally, linalool may exert its antimicrobial effects by disrupting biofilm formation [158,159,160]. Notably, while there is promising research on the antibacterial properties of linalool, more studies are needed to better understand its efficacy, mechanisms of action, and potential applications in clinical settings. Additionally, the concentration of linalool and the specific bacterial strains being targeted can influence its effectiveness.

#### 3.3.5. Limonene

Limonene, a cyclic monoterpene hydrocarbon, is commonly present in the peels of citrus fruits such as oranges, lemons, limes, and grapefruits, imparting their characteristic citrus aroma and flavor. Its versatile properties have led to widespread use across various industries, including food and beverage, cosmetics, cleaning products, and aromatherapy, owing to its pleasant fragrance and adaptable nature [161]. Although limonene is predominantly recognized for its aromatic attributes and its function as a solvent in cleaning solutions, it also demonstrates antibacterial properties. Studies indicate that limonene exhibits antimicrobial effects against diverse bacteria, encompassing both Gram-positive and Gram-negative strains. The mechanism involves the leakage of intracellular proteins, lipids, and nucleic acids, confirming membrane damage and disruption of cell permeability barriers. Additionally, the release of intracellular ATP further suggests disruption of the membrane barrier. Interaction with DNA underscores limonene’s ability to unwind plasmids, potentially inhibiting DNA transcription and translation processes [161,162].

The MIC and MBC of limonene have been established to be 10 and 12 μg/mL, respectively. Limonene exhibited the least antibacterial efficacy compared to other compounds tested, with an inhibition zone of 9 mm against EHEC bacteria [119]. Di Pasqua et al. (2006) determined alterations in membrane fatty acid composition induced by limonene. The concentration of unsaturated fatty acids (UFAs) notably increased when cells were exposed to limonene. This resulted from decreased palmitic acid levels and elevated concentrations of linoleaidic, docosanoic, and eicosapentaenoic acids [94]. Moreover, the anti-quorum-sensing potential of d-limonene in nano form against *E. coli* O157:H7 was investigated. The MIC was established at 5% (*v*/*v*), with sub-MIC levels of 2.5% (*v*/*v*) and 1.25% (*v*/*v*) selected for further evaluation. The study revealed that d-limonene nanoemulsion hindered *E. coli* biofilm formation by suppressing curli and extracellular polymeric substance production without impeding cell growth. Additionally, it reduced swimming and swarming abilities. Further analysis demonstrated interference with auto-inducer 2 (AI-2) communication by d-limonene nanoemulsion, which led to the repression of curli-related and AI-2 importer genes in *E. coli* [163]. Espina and Gelaw investigated the combined juice preservation techniques involving pulsed electric field (PEF) treatment and lethal heat supplemented with (+)-limonene against *E. coli* O157:H7. When applied in combination, the processes resulted in the inactivation of an additional 4.0 log or more compared to individual treatments. Consequently, this combination exhibited a synergistic effect on the overall inactivation. When (+)-limonene was applied separately at a concentration of 200 mL/L at 20 °C against an initial population of 3 × 10^7^ CFU/mL of *E. coli* O157:H7 suspended in juices for 10 min, less than 0.5 log of the initial population were inactivated. Conversely, a separate PEF treatment in the absence of (+)-limonene resulted in the inactivation of less than 0.5 log of the initial *E. coli* O157:H7 population and caused sub-lethal injury to the bacterial outer membrane in less than 1.0 log of surviving cells. The combined process involving PEF with (+)-limonene resulted in additive final inactivation levels, equivalent to the sum of the inactivation levels of both treatments applied separately. No additional inactivation was observed due to the simultaneous application of lethal heat treatment in the presence of (+)-limonene [164].

It is important to note that limonene efficacy can vary depending on factors such as concentration, bacterial strain, and environmental conditions. Further research is needed to fully understand the antimicrobial properties of limonene and its potential applications in clinical settings. Overall, limonene represents a natural compound with diverse biological activities, and ongoing studies continue to explore its therapeutic potential in various contexts, including antimicrobial and antibacterial applications.

#### 3.3.6. Other Terpenes and Terpenoids

Several other natural compounds exhibit significant antimicrobial properties against EHEC. Compounds such as terpineol, camphor, and pinene primarily disrupt bacterial cell membranes and interfere with metabolic functions, leading to bacterial eradication [133,165,166].

Terpineol, a monoterpene alcohol found in pine oil, cajuput oil, and eucalyptus oil, has a hydroxyl group that interacts with bacterial carbohydrates and other chemical groups [133,167]. This interaction leads to changes in phospholipid polar head groups and decreased membrane fluidity, disrupting the electron transport chain and causing a collapse of membrane potential and pH balance. At a concentration of 0.8%, α-terpineol reduces *E. coli* O157 populations by about 5.6 log CFU/mL within just one hour, attributed to ATP leakage and membrane disruption [133]. Similarly, camphor, extracted from the wood of the camphor tree (*Cinnamomum camphora* Ness et Eberm.) or synthesized from turpentine oil, disrupts bacterial cell membranes and interferes with metabolic processes [165,166]. It demonstrates significant antibacterial activity against the O157:H7 strain, with an MIC of 7.0 μg/mL and MBC of 8.0 μg/mL. A zone of growth inhibition measuring 13 mm was observed [119], indicating its effectiveness in hindering bacterial proliferation. However, not all studies have found camphor to be highly effective; for instance, when combined with high hydrostatic pressure (HHP), camphor achieved less than a 2.0 log reduction in bacterial counts, suggesting variable effectiveness depending on environmental conditions [130]. Pinene, a bicyclic monoterpene hydrocarbon found predominantly in coniferous trees, exists in two isomeric forms: α-pinene and β-pinene [168]. Both isomers inhibit microbial cell membrane integrity, exhibiting inhibition zones of 10 mm against the *E. coli* strain tested. They have similar MIC values of 8.0 μg/mL and MBCs of 10.0 μg/mL [119]. Despite their antibacterial properties, pinene isomers do not exhibit significant antioxidative activity [131].

In contrast, compounds like geraniol and isolimonic acid focus on inhibiting biofilm formation and bacterial adhesion, which are crucial for bacterial colonization and infection. Geraniol, found in the essential oils of roses, citronella, and geraniums [169], acts as an uncompetitive inhibitor of glucosyltransferase, an enzyme essential for biofilm formation. It reduces bacterial adhesion on stainless steel surfaces and decreases glucan production, with an MIC of 3.0 mg/mL against *E. coli* O157:H7. Docking analyses suggest that geraniol interacts with the helix finger of glucosyltransferase responsible for polymer production, thereby inhibiting biofilm formation [136]. Isolimonic acid, a triterpenoid compound isolated from certain pine trees, is a strong inhibitor of biofilm formation and attachment in EHEC [138,139,170]. It acts by suppressing the expression of LEE and flagellar operons, disrupting the type III secretion system. Isolimonic acid shows significant inhibition of biofilm formation, with an inhibitory concentration (IC25) of 19.7 μM, and has been shown to inhibit the formation and attachment of biofilms to Caco-2 cells. It is believed to exert its effects by inhibiting AI-3/epinephrine-mediated cell–cell signaling through the QseBC and QseA pathways, although the precise mechanisms remain to be fully elucidated [138].

Eucalyptol (1,8-cineole), commonly extracted from eucalyptus species like *Eucalyptus globulus* Labill. and *Ecklonia radiata* J. Agardh, as well as other aromatic plants like rosemary and tea tree, enhances its antimicrobial efficacy through high water solubility, facilitating diffusion into agar media and interaction with microbial cells [119,171,172]. It elongates the lag phase of bacterial growth, inhibiting proliferation over time [131]. Eucalyptol exhibits an inhibition zone of 18.0 mm against *E. coli* O157, with bacteriostatic activity observed at concentrations as low as 6.0 μg/mL (MIC) and bactericidal activity at 8.0 μg/mL (MBC) [119]. At a concentration of 0.6%, it has been found to elongate the lag phase of *E. coli* O157:H7, indicating its potential to inhibit bacterial growth over time. However, it shows weak antioxidant activity by the ferric reducing antioxidant power (FRAP) method and almost no free radical scavenging activity with the DPPH method [131].

These compounds, through their shared and unique mechanisms of action at specific concentrations, demonstrate the potential for developing effective natural antimicrobial agents. Oxygenated monoterpenes—eucalyptol, terpineol, geraniol, and camphor—generally exhibit higher antimicrobial potential compared to hydrocarbon monoterpenes like pinene. This increased efficacy is partly due to their higher water solubility, which facilitates diffusion through agar media and interaction with microbial cells [119]. Functional groups such as hydroxyl groups in terpineol and geraniol enhance their ability to form hydrogen bonds with microbial cell components, disrupting membrane integrity and metabolic processes [119,133,167]. These findings underscore the potential of natural compounds in developing new antimicrobial agents or natural preservatives for the food and pharmaceutical industries. The increasing consumer preference for natural alternatives to chemical bactericides, coupled with concerns about microbial resistance, positions these compounds as promising candidates for enhancing food safety and developing novel therapeutic applications. However, further research is necessary to fully understand their mechanisms of action, optimize their efficacy, and ensure their safety in practical applications.

## 4. Perspectives

With the rise of antibiotic resistance, plant-derived compounds—or phytochemicals—are increasingly recognized as promising alternatives and adjuncts for combating bacterial infections. Compounds like polyphenols, terpenes, and ITCs exhibit diverse antimicrobial mechanisms, including membrane disruption, inhibition of virulence signaling pathways, and suppression of genes crucial for bacterial pathogenicity (Figure 5). While certain phytochemicals, such as thymol and eucalyptol, are already FDA-approved for over-the-counter antiseptic products, others, like EGCg and sulforaphane, are under active investigation in clinical trials. Nevertheless, phytochemicals offer significant potential for integration into existing food-safety protocols as natural antimicrobials [173,174,175]. These compounds can serve as secondary hurdles, complementing established practices like pasteurization, refrigeration, and chemical preservatives, to provide additional protection against microbial contamination. Effective integration requires careful consideration of several factors, including their antimicrobial efficacy under typical storage and processing conditions, and their compatibility with various food matrices. Furthermore, compliance with regulatory frameworks is necessary, including safety evaluations, toxicological assessments, and adherence to labeling standards set by regulatory agencies such as the Food and Drugs Administration (FDA) and the European Food Safety Authority (EFSA). This review underscores the potential of these bioactive compounds to modulate EHEC virulence specifically and highlights how their unique mechanisms can effectively support or potentially replace traditional antibiotics. As research progresses, the distinct properties of phytochemicals may offer safer and more targeted solutions for controlling bacterial pathogens without fostering resistance.

### Challenges and Future Research Directions

For phytochemicals to move effectively from bench to bedside, several challenges must be addressed:**Integrating phytochemicals into existing food safety protocols or regulatory frameworks:** Effective incorporation of phytochemicals requires establishing their role in promoting functional foods or “superfoods”, and standardizing food products for consistent phytochemical quality, content, and aligning dietary recommendations. Phytochemicals can serve as alternative strategies to antibiotic treatments, supporting a holistic One Health approach that addresses human, animal, and environmental health while minimizing chemical use.**Bioavailability and Targeted Delivery:** Ensuring that phytochemicals maintain stability and bioactivity in the gastrointestinal environment is critical for controlling EHEC effectively. Advanced delivery systems, such as encapsulation or nanoformulations, could enhance bioavailability and direct these compounds to targeted sites within the gut.**Clinical Trials for EHEC-Specific Applications:** While preclinical data are promising, rigorous clinical studies focused explicitly on EHEC are essential to confirm the safety, efficacy, and optimal dosages of these compounds in human populations. Current evidence from applications beyond EHEC provides a strong foundation, but EHEC-specific trials will be necessary to substantiate therapeutic potential.**Interactions with the Gut Microbiome:** Understanding how phytochemicals influence the host microbiome is crucial to avoid unintended disruption of beneficial gut flora. Studying these interactions can help ensure that treatment selectively targets EHEC while maintaining the balance of the broader microbial community.**Interactions with Other Therapeutic Agents:** Phytochemicals may enhance or interfere with conventional antibiotics and other drugs. Research should investigate these interactions to optimize combinations, aiming to boost efficacy and reduce potential conflicts in treatment.**Green-Synthesized Materials:** Secondary plant metabolites are utilized as biological precursors and active components of new nanomaterials such as metal oxides, delivery systems, bio-sensors, or antiseptic fabrics.

## 5. Conclusions

This review contributes to the ongoing search for innovative strategies against EHEC, especially given the limitations of antibiotics for treating this infection. Phytochemical-based therapies, grounded in a targeting of EHEC virulence mechanisms, offer an alternative to traditional antibiotics and may lead to more sustainable, less resistance-prone treatments. Moving forward, the development of these plant-derived agents as therapeutic interventions should be prioritized, not only for EHEC but potentially also for a broader range of bacterial infections where antibiotic resistance is a concern.

## Figures and Tables

**Figure 1 ijms-26-00381-f001:**
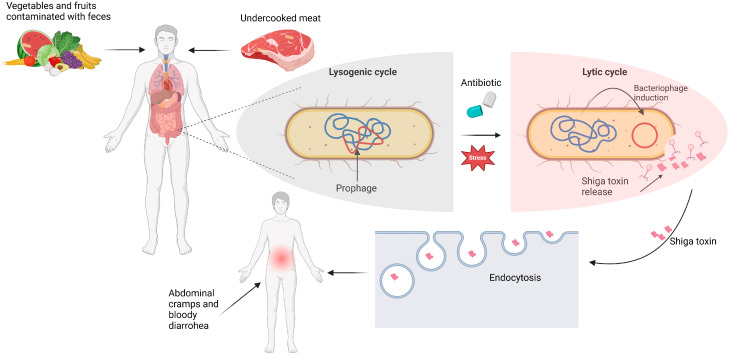
Mechanism of enterohemorrhagic *Escherichia coli* (EHEC) infection and Shiga toxin production. The scheme outlines the infection pathway following the ingestion of contaminated food, such as vegetables, fruits, or undercooked meat. EHEC carries a prophage that remains latent during the lysogenic cycle. Upon exposure to environmental stressors, including antibiotic treatment, the prophage is activated, triggering the lytic cycle and subsequent Shiga toxin production. The toxin enters host cells through endocytosis, leading to severe gastrointestinal symptoms, including abdominal cramps and bloody diarrhea. The presented scheme was prepared based on [4,5]. Created in BioRender. https://BioRender.com/y08b332.

**Figure 2 ijms-26-00381-f002:**
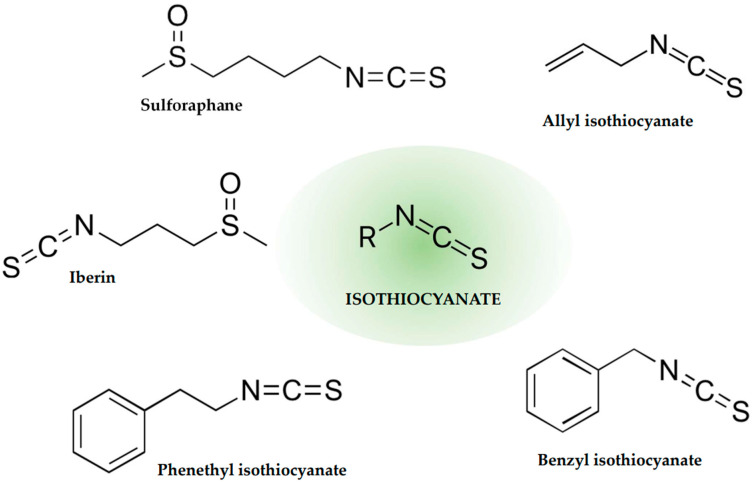
Structural representation of isothiocyanates (R–N=C=S) and their key derivatives. In green shade, the general structure of ITC is presented.

**Figure 3 ijms-26-00381-f003:**
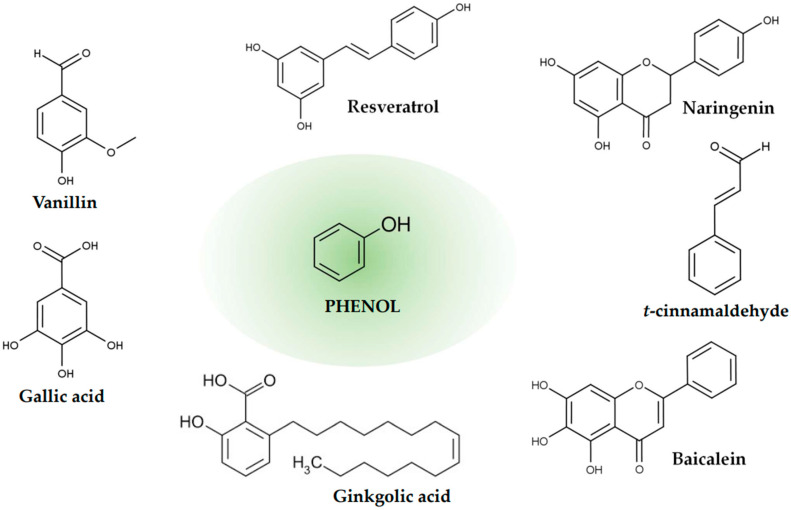
Structural representation of phenolic compounds and their derivatives. The central phenol structure (green-shaded area) highlights the core functional group present in all depicted molecules.

**Figure 4 ijms-26-00381-f004:**
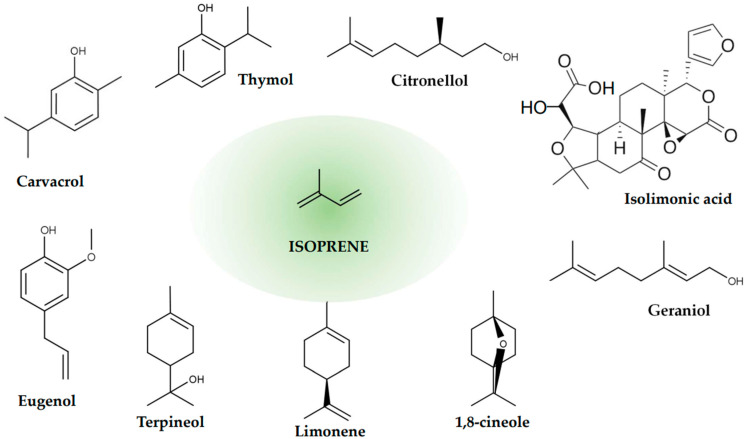
Structural representation of plant-derived terpene derivatives. The central isoprene unit (highlighted in green) serves as the core building block of terpenes and terpenoids.

**Figure 5 ijms-26-00381-f005:**
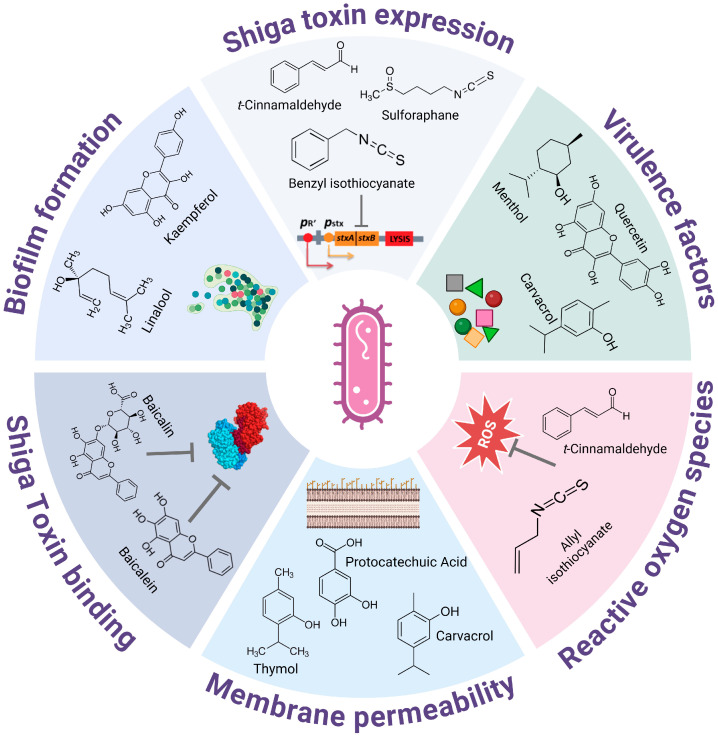
Representation of natural compounds and their targeted actions against enterohemorrhagic *E. coli* (EHEC) pathogenesis. The highlighted pathways show the compounds’ potential to reduce bacterial virulence, toxin production, and ROS generation, while enhancing membrane disruption and biofilm inhibition. Created in BioRender. https://BioRender.com/q50t463.

**Table 1 ijms-26-00381-t001:** Anti-EHEC activity of phenolic compounds.

Compound Type	Compound	Origin	Concentration	Mechanism/Effects on EHEC	Reference
Flavonoids	Apigenin	Citrus fruit	Not specified	Inhibits biofilm formation by suppressing QS signals	[61]
	Baicalein/baicalin	Chinese skullcap (*Scutellaria baicalensis* Georgi)	0.13 mmol/L	Inhibits Stx2 oligomer formation, reducing cytotoxicity and renal damage; suppresses virulence gene expression.	[62,63,64]
	Quercetin	Caper (*Capparis* sp.), onions (*Allium cepa* L.), kale (*Brassica oleracea acephala*)	≤1.4 g/L 32 μg/mL	Suppresses biofilm formation and virulence through inhibition of *luxS* and *pfs*	[61,65,66]
	Kaempferol	Citrus fruit, various plants	100 μg/mL 32 μg/mL	Inhibits growth and biofilm formation by QS modulation	[61]
	Naringenin	Citrus fruits	7.3–11.0 mM	Inhibits biofilm formation, nonspecific quorum-sensing inhibitor	[61,67]
	Chrysoeriol, sudachitin, acacetin	*Chrysanthemum lavandulifolium*	10 mg/mL	Downregulates *stx1A* and *stx1B* gene expression, reducing toxin synthesis	[68]
	Catechins (e.g., epigallocatechin)	Green tea (*Camellia sinensis* L.)	5 mg/mL	Reduced *ompC* and *rpoS* expression, inhibition of ROS dysmutase activity	[69,70,71]
Phenolic acids	Gallic acid	Grains, tea (*Camellia sinensis* L.)	1.0–5 mg/mL	Inhibits swarming motility, biofilm formation, and quorum sensing (*csgA*, *fimA*, *eae* genes)	[72,73,74]
	Methyl gallate	Not specified	0.02–0.4 mg/mL	Overexpressed *csgA* and *cyaA* without increasing biofilm, reduced motility by 41.8% at 0.05 mg/mL	[73]
	Protocatechuic acid	Various plants	MIC 2.5 mg/mL	Disrupted cell membranes, increasing permeability, synergistic with high hydrostatic pressure	[75]
	Vanillic acid	Grains, fruits, herbs	10 mM	5.0 log reduction in bacterial populations after 7 days	[76]
	Caffeic acid	Apple (*Malus domestica* Borkh.)	0.2 mg/mL	Reduced viable counts below detection levels at pH 3.2	[77]
	p-Coumaric acid	Peanuts (*Arachis hypogaea* L.), beans (*Phaseolus vulgaris* L.), tomatoes (*Solanum lycopersicum* L.)	0.5%	Increased *E. coli* O157:H7 death rate by 23-fold	[78]
Phenylopropanoids	Coumarin and derivatives	Cinnamon (*Cinnamomum verum* J. Presl), mint (*Mentha × piperita* L.), green tea (*Camellia sinensis* L. Kuntze), lavender (*Lavandula angustifolia* Mill.)	50 μg/mL	Inhibited biofilm formation >80%, repressed curli and motility genes, esculetin repressed Shiga-like toxin II gene	[79,80]
	*t*-Cinnamaldehyde	Cinnamon (*Cinnamomum verum* J. Presl, *Cinnamomum cassia* L. J. Presl)	MIC 1.875–3.750 mM	Inhibition of bacterial growth by the induction of the stringent response, blocking of phage lytic cycle and Shiga toxin production	[81,82,83,84,85,86]
	Ginkgolic acid	Ginkgo (*Ginkgo biloba* L.)	1–5 μg/mL	Inhibited biofilm formation of up to 70% at 1 μg/mL and nearly abolished at 5 μg/mL, reduced fimbriae, altered motility, suppressed curli expression	[87,88]
	β-Resorcylic Acid	Various angiosperms	1%	Reduces bacterial population on surfaces by disrupting biofilm formation and cell adhesion	[89]
Stilbens	Resveratrol and derivatives	Grapes (*Vitis vinifera* L.), red wine, peanuts (*Arachis hypogaea* L.),	8.7–13.0 mM	Represses motility and biofilm-related genes (*flh*, *csg*, *fim*, *motB*), inhibits QS pathways	[67,90,91,92]
Other	Eugenol	Basil (*Ocimum basilicum* L.), cinnamon (*Cinnamomum verum* J. Presl), cloves (*Syzygium aromaticum* (L.) Merr. & Perry)	0.5% *w*/*w* in food preservation	Increases membrane permeability, inhibits biofilm formation through downregulation of *csg* and *fim* genes	[93,94,95,96,97]
	Phloretin	Apple (*Malus domestica* Borkh.)	25–50 μg/mL	Reduced biofilm formation; repressed toxin genes (e.g., *hlyE*, *stx2*), autoinducer-2 importer genes, curli genes; reduced fimbriae production	[98]
	Propolin D	Parasol leaf tree (*Macaranga tanarius* (L.) Müll.Arg.)	10 to 50 µg/ml	Disrupts biofilm integrity by inhibiting *csgA* and *csgB* expression, reducing curli production	[99]
	Total blueberry phenolics	Wild blueberries (*Vaccinium myrtillus* L.)	0.8–1.8 g/L	Increased cell membrane permeability, leading to CFU reductions of 6.0–7.0 log for various fractions like MPA, A&P, and anthocyanins	[100]

**Table 2 ijms-26-00381-t002:** Plant-derived terpenes and terpenoids targeting EHEC virulence factors.

Compound	Origin	Concentration	Mechanism/Effects on EHEC	Reference
Carvacrol	Oregano (*Origanum vulgare* L.), thyme (*Thymus vulgaris* L.)	MIC 0.0005–0.310 mg/mL	Disrupts cell membranes, interferes with enzyme systems, downregulates virulence genes, inhibits biofilm formation	[40,94,117,118,119,120,121,122,123,124]
Thymol	Oregano (*Origanum vulgare* L.), thyme (*Thymus vulgaris* L.)	MIC 0.0001–0.310 mg/mL	Disrupts bacterial membranes, reduces motility, decreases biofilm formation, downregulates virulence genes	[94,118,119,121,125,126,127]
Menthol	Mint (*Mentha piperita* L., *Mentha spicata* L.)	MIC 0.0001–11 mg/mL	Disrupts bacterial cell membranes, reduces biofilm formation, decreases expression of virulence gene	[119,128,129]
Camphor	Cinnamon (*Cinnamomum camphora Ness et* Eberm.), thyme (*Thymus vulgaris* L.), lavender (*Lavandula angustifolia* Mill.), basil (*Ocimum basilicum* L.), sage (*Salvia officinalis* L.)	MIC 0.008 mg/mL	Disrupts cell membranes, interferes with metabolic functions	[119,130]
Alpha/beta-pinene	Mint (*Mentha spicata* L.), lemon (*Citrus limon* Burm.), sage (*Salvia officinalis* L.), thyme (*Thymus vulgaris* L.)	MIC 0.008 mg/mL	Inhibits microbial cell membrane integrity	[118,119]
Linalool	Lavender (*Lavandula angustifolia* Mill.), basil (*Ocimum basilicum* L.), mint (*Mentha piperita* L.), chamomile (*Matricaria chamomilla* L.), thyme (*Thymus vulgaris* L.)	MIC 0.006 mg/mL	Disrupts bacterial cell membranes, interferes with membrane permeability, inhibits bacterial enzymes, disrupts biofilm formation	[119]
Eucalyptol (1,8-cineole)	Mint (*Mentha spicata* L., *Mentha piperita* L.), chamomille (*Matricaria chamomilla* L.), lavender (*Lavandula angustifolia* Mill.), basil (*Ocimum basilicum* L.), sage (*Salvia officinalis* L.), thyme (*Thymus vulgaris* L.)	MIC 0.006–0.032 mg/mL	High water solubility enhances diffusion in agar media, elongates bacterial lag phase	[119,131,132]
α-terpineol	White jade orchid tree (*Michelia* × *alba* DC.)	MIC 2 mg/mL	Disrupts bacterial cell membranes, affects ATP leakage and membrane fluidity	[133,134,135]
Geraniol	Basil (*Ocimum basilicum* L.)	MIC 3 mg/mL	Inhibits glucosyltransferase, affecting biofilm formation	[136,137]
Limonene	Mint (*Mentha spicata* L., *Mentha piperita* L.), lemon (*Citrus limon* Burm., *Citrus aurantium* L.), chamomile (*Matricaria chamomilla* L.), lavender (*Lavandula angustifolia* Mill.), basil (*Ocimum basilicum* L.), sage (*Salvia officinalis* L.), thyme (*Thymus vulgaris* L.)	MIC 0.01 mg/mL	Disrupts cell membranes, causes leakage of intracellular contents, interacts with DNA, unwinds plasmids	[94,119,134]
Isolimonic acid	Bitter orange (*Citrus* × *aurantium* L.)	IC25 19.7 μM	Inhibits biofilm formation and attachment, disrupts TTSS and AI-3/epinephrine signaling	[138,139]

## Data Availability

Not applicable.

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
