# Peer review of "Phytochemicals Controlling Enterohemorrhagic Escherichia coli (EHEC) Virulence—Current Knowledge of Their Mechanisms of Action"

_ijms, 2025, doi:10.3390/ijms26010381_

Round 1

Reviewer 1 Report

Comments and Suggestions for Authors

I found your work as interesting and important. However, I have some comments. (1) I suggest to include into MS an information on the literature searching (period of publication, databases, key words, inclusion/exclusion criteria, etc.). (2) Latin plant names usually are composed of 3 elements (genus, species, abbreviation of person described a plant, eg. Thymus vulgaris L.). (3) p. Would you like to explain a differentiation between phenols and polyphenols? (4) I suggest to modify key words by removing "Phytochemicals" and "Antibiotic resistance". You can include e.g. "Shiga toxin" as main virulence factor (only "Biofilm" was included).

Author Response

I found your work as interesting and important. However, I have some comments.

We thank the reviewer for their thoughtful evaluation and valuable comments. Below, we address each point in detail (in blue):

(1) I suggest to include into MS an information on the literature searching (period of publication, databases, key words, inclusion/exclusion criteria, etc.).

Thank you for the suggestion. Specifically, we used databases such as PubMed and Web of Science. However, the manuscript was built as descriptive review, the qualitative search depended on analysing the content of a given studies. For full coverage of the publication base in the topic, we did not assume time restrictions.

(2) Latin plant names usually are composed of 3 elements (genus, species, abbreviation of person described a plant, eg. Thymus vulgaris L.).

Amended and done.

(3) p. Would you like to explain a differentiation between phenols and polyphenols?

More comprehensive explanation of the compound’s nomenclature has been provided to MS (lines- 334-337)

(4) I suggest to modify key words by removing "Phytochemicals" and "Antibiotic resistance". You can include e.g. "Shiga toxin" as main virulence factor (only "Biofilm" was included).

Amended and done

Reviewer 2 Report

Comments and Suggestions for Authors

The review article titled "Phytochemicals controlling enterohemorrhagic Escherichia coli (EHEC) virulence – current knowledge of their mechanisms of action" discusses the virulence of Enterohemorrhagic Escherichia coli (EHEC), a common foodborne pathogen, and the potential of phytochemicals as antivirulence agents. The main points presented in the article are the following:

EHEC Virulence Factors:

Shiga Toxin Production: EHEC produces Shiga toxins, which can lead to severe complications like haemolytic-uraemic syndrome (HUS).

Locus of Enterocyte Effacement (LEE): EHEC uses a type 3 secretion system to inject effector proteins into host cells, aiding in colonization and infection.

Biofilm Formation: EHEC forms biofilms, which help it survive under unfavourable conditions and enhance its virulence.

Phytochemicals with Antimicrobial Activity Against EHEC:

Isothiocyanates (ITCs): These compounds in plants like broccoli and mustard inhibit EHEC growth and virulence by inducing a stringent response.

Phenolic Compounds: Flavonoids, phenolic acids, and other phenolics inhibit EHEC biofilm formation and toxin production.

Terpenes and Terpenoids: Compounds like carvacrol, thymol, and menthol disrupt EHEC cell membranes and inhibit biofilm formation.

Mechanisms of Action:

Stringent Response: ITCs induce a stress response in EHEC, inhibiting its growth and toxin production.

Membrane Disruption: Many phytochemicals disrupt EHEC cell membranes, leading to cell death.

Inhibition of Virulence Genes: Phytochemicals suppress the expression of genes involved in EHEC virulence.

The document highlights the potential of phytochemicals as alternative or complementary therapies to antibiotics for controlling EHEC infections, emphasizing the need for further research to fully understand their mechanisms and optimize their use.

The manuscript is engaging and thoughtfully organized, featuring well-designed tables and figures, and is suitable for publication in its present form.

Author Response

The review article titled "Phytochemicals controlling enterohemorrhagic Escherichia coli (EHEC) virulence – current knowledge of their mechanisms of action" discusses the virulence of Enterohemorrhagic Escherichia coli (EHEC), a common foodborne pathogen, and the potential of phytochemicals as antivirulence agents. The main points presented in the article are the following:

EHEC Virulence Factors:

Shiga Toxin Production: EHEC produces Shiga toxins, which can lead to severe complications like haemolytic-uraemic syndrome (HUS).

Locus of Enterocyte Effacement (LEE): EHEC uses a type 3 secretion system to inject effector proteins into host cells, aiding in colonization and infection.

Biofilm Formation: EHEC forms biofilms, which help it survive under unfavourable conditions and enhance its virulence.

Phytochemicals with Antimicrobial Activity Against EHEC:

Isothiocyanates (ITCs): These compounds in plants like broccoli and mustard inhibit EHEC growth and virulence by inducing a stringent response.

Phenolic Compounds: Flavonoids, phenolic acids, and other phenolics inhibit EHEC biofilm formation and toxin production.

Terpenes and Terpenoids: Compounds like carvacrol, thymol, and menthol disrupt EHEC cell membranes and inhibit biofilm formation.

Mechanisms of Action:

Stringent Response: ITCs induce a stress response in EHEC, inhibiting its growth and toxin production.

Membrane Disruption: Many phytochemicals disrupt EHEC cell membranes, leading to cell death.

Inhibition of Virulence Genes: Phytochemicals suppress the expression of genes involved in EHEC virulence.

The document highlights the potential of phytochemicals as alternative or complementary therapies to antibiotics for controlling EHEC infections, emphasizing the need for further research to fully understand their mechanisms and optimize their use.

The manuscript is engaging and thoughtfully organized, featuring well-designed tables and figures, and is suitable for publication in its present form.

We thank the Reviewer for the detailed analysis of the manuscript content and appreciation of our work.

Reviewer 3 Report

Comments and Suggestions for Authors

Author reported “Phytochemicals Controlling Enterohemorrhagic Escherichia coli (EHEC) Virulence – Current Knowledge of Their Mechanisms of Action" provides a detailed review of plant-derived phytochemicals and their potential as antivirulence agents against EHEC. The manuscript highlights various phytochemicals, including isothiocyanates, phenolic compounds, and terpenes, focusing on their mechanisms of action and effects on EHEC virulence factors. The article can be accepted but I suggest author should consider following comments to improve the article

  1. I suggest the author provide a stronger justification for focusing on antivirulence strategies, specifically highlighting their advantages over conventional antibiotic approaches.

2.     Consider adding a specific section on the role of metal oxides in inhibiting E. coli, as they are often significant in antibacterial studies and could strengthen overall discussion.

Novel Nd-N/TiO2 Nanoparticles for Photocatalytic and Antioxidant Applications Using Hydrothermal Approach Materials 2022, 15(19), 6658:

Phyto Synthesis of Manganese-Doped Zinc Nanoparticles Using Carica papaya Leaves: Structural Properties and Its Evaluation for Catalytic, Antibacterial and Antioxidant Activities: Polymers 2022, 14(9), 1827

  1. Add some reference for the proposed mechanism of infection depicted in Figure 1 with proper explination.
  2. Author should specify negative controls used in testing the efficacy of phytochemicals against EHEC and compare their results.
  3. Explain briefly why specific phytochemicals like cinnamaldehyde and isothiocyanates are important.
  4. Provide a more detailed explanation of how cinnamaldehyde interacts with the bacterial cellular machinery to inhibit Shiga toxin production.
  5. Include concentration ranges or minimal inhibitory concentrations in Figure 2.
  6. Explain  how these phytochemicals could be integrated into existing food safety protocols or regulatory frameworks.

Author Response

Author reported “Phytochemicals Controlling Enterohemorrhagic Escherichia coli (EHEC) Virulence – Current Knowledge of Their Mechanisms of Action" provides a detailed review of plant-derived phytochemicals and their potential as antivirulence agents against EHEC. The manuscript highlights various phytochemicals, including isothiocyanates, phenolic compounds, and terpenes, focusing on their mechanisms of action and effects on EHEC virulence factors. The article can be accepted but I suggest author should consider following comments to improve the article

We thank the Reviewer for the thorough evaluation and valuable comments that have significantly improved the quality of our manuscript. Below, we address each comment in detail (in blue):

  1. I suggest the author provide a stronger justification for focusing on antivirulence strategies, specifically highlighting their advantages over conventional antibiotic approaches.

Thank you for your suggestion. Now the more elaborate discussion under justification of searching novel therapeutic strategies is added. We also highlighted the supportive therapy availability for EHEC treatment, and that antibiotics worsen the symptoms of the disease. (Lines – 65-67; 69-70; 944)

  1. Consider adding a specific section on the role of metal oxides in inhibiting E. coli, as they are often significant in antibacterial studies and could strengthen overall discussion.

Novel Nd-N/TiO2 Nanoparticles for Photocatalytic and Antioxidant Applications Using Hydrothermal Approach Materials 2022, 15(19), 6658:

Phyto Synthesis of Manganese-Doped Zinc Nanoparticles Using Carica papaya Leaves: Structural Properties and Its Evaluation for Catalytic, Antibacterial and Antioxidant Activities: Polymers 2022, 14(9), 1827

Thank you for pointing this out, now the more comprehensive discussion was added to the Perspectives section (Lines – 938-940)

  1. Add some reference for the proposed mechanism of infection depicted in Figure 1 with proper explination.

Thank you for the suggestion. We have now added appropriate references to the legend of Figure 1

  1. Author should specify negative controls used in testing the efficacy of phytochemicals against EHEC and compare their results.

Thank you for this valuable comment. In the EHEC model, the activation of the SOS response and the subsequent production of bacteriophage-encoded toxins are critical. Most EHEC infections are asymptomatic or do not lead to severe complications, making it essential to investigate factors that activate these mechanisms, such as DNA breaks or ROS generation. Antibiotics disrupting nucleic acid synthesis or inducing ROS are typically used as controls (doi: 10.1128/AAC.02159-19). Additionally, genetically modified strains lacking key regulatory factors involved in bacteriophage/toxin synthesis serve as negative controls (doi: 10.3390/toxins13080534; doi: 10.1128/AAC.00958-21). Promising phytochemical candidates should meet these criteria and undergo cytotoxicity testing within the concentration ranges inhibiting bacterial growth. We also noted that responses to antibiotics can vary significantly depending on the strain, necessitating the use of multiple strains in studies. However, unifying results across studies remains challenging, underscoring the need for standardized, top-down criteria.

  1. Explain briefly why specific phytochemicals like cinnamaldehyde and isothiocyanates are important.

We appreciate the comment. We have made an effort to highlight all crucial aspects of compounds health benefits and important traits. Now the description of the agents has been improved where applicable. (Lines- 189-205; 442-444)

  1. Provide a more detailed explanation of how cinnamaldehyde interacts with the bacterial cellular machinery to inhibit Shiga toxin production.

We have provided a more detailed explanation of how cinnamaldehyde interferes with bacterial cellular processes to inhibit Shiga toxin production. These details have been added to the relevant section. (Lines- 454-58)

  1. Include concentration ranges or minimal inhibitory concentrations in Figure 2.

Thank you to highlight this issue. To maintain a consistent figure style, we have added the concentration ranges and minimal inhibitory concentrations (MICs) in the accompanying text. (Lines – 255-56; 270-71; 279-80)

  1. Explain  how these phytochemicals could be integrated into existing food safety protocols or regulatory frameworks.

Thank you for your comment which improved our manuscript. We added specific description into the Perspectives section (Lines-916-19)

Round 2

Reviewer 3 Report

Comments and Suggestions for Authors

The response letter is not clear. Most of the comments are unanswered . or very short response where provided. I request the author to crafully address the following comments; 

1. I suggest the author provide a stronger justification for focusing on antivirulence strategies, specifically highlighting their advantages over conventional antibiotic approaches. 

2.     Consider adding a specific section on the role of metal oxides in inhibiting E. coli, as they are often significant in antibacterial studies and could strengthen overall discussion. Novel Nd-N/TiO2 Nanoparticles for Photocatalytic and Antioxidant Applications Using Hydrothermal Approach Materials 2022, 15(19), 6658: Phyto Synthesis of Manganese-Doped Zinc Nanoparticles Using Carica papaya Leaves: Structural Properties and Its Evaluation for Catalytic, Antibacterial and Antioxidant Activities: Polymers 2022, 14(9), 1827

3. Add some reference for the proposed mechanism of infection depicted in Figure 1 with proper explination.

4. Explain briefly why specific phytochemicals like cinnamaldehyde and isothiocyanates are important.

5. Provide a more detailed explanation of how cinnamaldehyde interacts with the bacterial cellular machinery to inhibit Shiga toxin production.

6. Explain  how these phytochemicals could be integrated into existing food safety protocols or regulatory frameworks.

Author Response

We would like to thank the esteemed Reviewer for the detailed and critical opinion, which enabled us to significantly improve the submitted manuscript. In the new version, we have significantly increased the suggested content descriptions. We have added changes to the previous version (saved in the option “track changes”).  The newest corrections are highlighted in yellow. We hope that all requirements have now been met and that all gaps in the consistency of the text have been closed.

Round 3

Reviewer 3 Report

Comments and Suggestions for Authors

Article can b accepted